# The Temporary Agency Worker’s Motivation Profile Analysis

**DOI:** 10.3390/ijerph18136779

**Published:** 2021-06-24

**Authors:** Filipa Sobral, Maria José Chambel, Filipa Castanheira

**Affiliations:** 1Research Centre for Human Development, Faculdade de Educação e Psicologia, Universidade Católica Portuguesa, 4169-005 Porto, Portugal; 2Faculdade de Psicologia, Universidade de Lisboa, 1649-013 Lisboa, Portugal; mjchambel@psicologia.ulisboa.pt; 3Nova School of Business and Economics, Universidade Nova de Lisboa, 2775-405 Carcavelos, Portugal; fcastanheira@novasbe.pt

**Keywords:** motivation, commitment, human resources management, cross sectional studies

## Abstract

The Self-Determination Theory (SDT) establishes that human motivations can take different forms (e.g., amotivation, extrinsic and intrinsic motivation), yet it is only recently that the theory has been advanced to explain how these different forms combine to influence temporary agency workers’ (TAWs) affective commitment and their perception over the human resources practices (HRP) applied. We tested this theory with data from seven temporary agency companies (N = 3766). Through latent profile analysis (LPA) we identified five distinct motivation profiles and found that they differed in their affective commitment to the agency and to the client-company, and in their perception of HRP. We verified that temporary agency workers in more intrinsic profiles had more positive outcomes and a better perception of the investment made by the companies, than did TAWs in more extrinsic profiles. Additionally, when TAWs were able to integrate the reasons for being in this work arrangement, the negative effect of the extrinsic motivation was attenuated, and it was possible to find moderated profiles in which TAWs also showed more positive results than TAWs with only extrinsic motives. These differences are consistent with the notion that a motivation profile provides a context that determines how the individual components are experienced. Theoretical and practical implications of this context effect are discussed.

## 1. Introduction

Today’s organization competitiveness depends on effectiveness, versatility, and the ability to respond to customers [1]. As a consequence, flexibility [2,3], along with the growth of contingent work (Organization for Economic Cooperation and Development [4,5], have become key concepts in work relations. It is increasingly common to find temporary workers side-by-side with those who are employed [3,6]. It is therefore crucial for companies to understand how to engage these workers jointly [7]. As mentioned by Mas and Pallais [8], the equilibrium between the firm demands, the worker preferences and the different possibilities of work arrangements is still an open question.

The study of temporary workers’ motivations, which is often connected with their voluntarism toward their employment situation, has persuasively demonstrated the existence of different reasons to accept this work arrangement [9,10]. However, these investigations either consider one kind of motivation alone or consider several motivations, but independently of each other. Considering the different ways people act, the Self-Determination Theory (SDT) [11] accounts for several types of motivation and connections between each motivation type and human learning, performance, and well-being [11]. According to SDT it is possible to consider: (a) intrinsic motivation, which means doing something for its own sake, and reflects the individual’s disposition to be challenged, to explore, and to develop social or cognitive competences; (b) extrinsic motivation, which is experienced in situations in which the individual acts more in accordance with the external regulation than in accordance with an intrinsic interest; and (c) amotivation, which relates to a state of total indifference [11,12].

Based on SDT, Meyer [13] underscored several findings on human motivation profiles, which have demonstrated that workers’ motivational states (i.e., intrinsic or extrinsic) are not mutually exclusive. It is evident that the same worker may present, simultaneously, both intrinsic and extrinsic motives in a specific work situation. De Jong, et al. [9] also confirmed that temporary workers could experience different forms of motivations at the same time. In opposition to the restrictiveness of a dichotomous analysis between intrinsic/voluntary and extrinsic/involuntary motives to be in this employment arrangement [14], the authors defend that a comprehensive study of temporary workers’ motivations should adopt a person-centered methodology and identify different motivation profiles.

Our research builds on the work of De Jong et al. [9] but upgrades its conclusions and overcomes previous limitations. First, De Jong and colleagues [9] considered different kinds of temporary employment in their sample. However, this may have a confounding effect because it is argued that the inconsistency across studies on temporary workers might derive from the sample characteristics, as these are sometimes aggregates of different temporary arrangements [15]. With this in mind, we use only temporary agency workers (TAWs). Of all the contingent work formats, temporary agency work is pointed to as the most complex work arrangement [15,16]. TAWs develop a double work relationship: one pertaining to the entity with whom they have a formal contract (i.e., the temporary agency) and another with the people with whom they actually work (i.e., the client-company) [17,18], an arrangement that might create “dual allegiance” issues [19]. Likewise, it is fundamental that academia adjusts and adapts the theoretical representations of labor relations to the present context [7,20].

Second, De Jong et al. [9] related the temporary workers’ motivation profiles with individual characteristics (i.e., age, gender, and family situation) and work factors (i.e., contract type, educational level, occupational position, tenure, employability, and work involvement) whereas our study relates TAWs’ motivation profiles with their employment relationship. Namely, our study differentiates the employment relationship established among TAWs, the agency, and the client company. In doing so, we offer an innovation by measuring TAWs’ affective commitment with the agency and the client-company. By reflecting the emotional liaison and identification that workers are able to establish with the organizations’ values and goals, affective commitment is identified as one of the most important variables related to favorable behavioral outcomes in workers [21]. It is therefore important to understand if TAWs’ commitments are related to their motivation to be in their current job situation, and consequently, to their overall employment relationship.

Additionally, we analyze the connection between TAWs’ motivations and their perception of the human resources practices (HRP). It is generally accepted that by sending certain messages, which may take the form of an HRP system, companies can connect with workers, emphasizing their importance and persuading them to respond in the same way [22,23]. However, the reactions to the HRP are not necessarily influenced by the fact of their existence [23]. The effect of actual HRP resides mostly in the perceptions they cause to the employee [23], and ultimately it is the fit between this investment and the individual characteristics that determines the effectiveness of a specific set of practices [24]. Hence, the HRP that companies implement may play a crucial role in responding to TAWs’ specific needs, which may influence attitudes capable of increasing workers’ motivation toward their employment situation.

Finally, the measure of motivations adopted in De Jong and colleagues’ [9] study can also be considered as a limitation. The authors themselves acknowledge that the use of just three motivations—voluntary, involuntary, and stepping-stone—each measured by one item, represents a restriction. Seeking to overcome this limitation, we distinguish four motivations—intrinsic, integrated, identified, and extrinsic—which are assessed through an adaptation toward TAWs of the Motivation at Work Scale [25].

### 1.1. Human Motivations: A Self-Determination Theory Perspective

In the SDT literature, the variation between autonomous/intrinsic and controlled/extrinsic motivation relates to the process of internalization—i.e., a natural inclination toward the integration of the norms and values of the social environment as part of the individual self [26]. Consequently, the extrinsic motivation covers the continuum between amotivation and intrinsic motivation, thus allowing individuals to incorporate external regulation as a hierarchical process that can occur by: (a) an integrated regulation, through which individuals are able to transform external regulation into self-regulation; (b) an identified regulation, through which individuals are able to act in accordance with external regulation, since they see it as congruently aligned with their goals and personal identity; (c) an introjected regulation, in which, although there is no full acceptance of the regulation by the self, individuals act as moved by their ego, wishing to maintain their self-worth; or (d) an external regulation, which represents the less intrinsic acceptance of the regulation, and individuals are moved to act only by instrumental reasons, such as through obtaining material reward or receiving reassurance of their personal purposes [12,27].

Relating this conceptualization to TAWs’ motivations, they can act: (a) intrinsically, making their choice voluntarily [10]; (b) extrinsically, moved by instrumental reasons, such as obtaining material reward or ensuring their personal purposes [12,27]—external regulation; or (c) they can be in a process of internalization and transform the external regulation into a more self-regulated choice. If so, they can (a) accept this arrangement because it fits their personal goals [28,29], allowing them to feel engaged in the situation because it is in line with who they are (i.e., accepting a temporary job because it is in accordance with their desire to be continuously challenged)—integrated regulation; (b) recognize the value of their job because it increases the opportunity for skills development and/or the chances to obtain other employment (e.g., stepping stone motive) [9]—identified regulation; or (c) feel that work is central to their life and believe that one must overcome the difficulties faced and be active in the labor market—introjected regulation. However, as argued by Lopes and Chambel [25], because temporary work is usually characterized by less job security [29,30], there are no differences between the introjected and the external motives. In fact, the impact of external regulations (e.g., the need to work for the money obtained or to “survive”) or introjected regulations (e.g., to avoid unemployment or the belief that being employed is fundamental to one’s life) should be approximately the same for TAWs. For this reason, the authors Lopes and Chambel [25] chose to include the introjected and external motivation jointly under the same label: external regulation (i.e., the job is only a way of surviving and earning some income—external regulation). Since we will be using their scale in our research, we also chose to use only the external regulation label for these two motivations.

#### Motivation Profiles Research

When looking at the SDT research that examines motivation profiles, the central argument is that when facing a certain situation, individuals may display both types of motivation, simultaneously—intrinsic and extrinsic [31,32,33]. As such, it is possible to create different paths of response, by conjugating their motivations differently. In order to detect these unobserved similarities and indicate groups that are homogeneous in their responses, it is necessary to use an adequate methodology, such as the Latent Profile Analysis (LPA) [34]. As it is a personal-centered technique, the LPA sets the focus on the relationships among the individuals, making it possible to sort groups of individuals who are similar to each other and different from the individuals in other groups [35].

By applying theoretical comparisons using a between-group analysis, the LPA methodology assumes the principles of a quasi-experimentation and allows stronger inferences about causality [36]. As such, this analysis allows us to define profiles in accordance with the specific combination of different motivations (i.e., more intrinsic or more extrinsic) for each individual [35] and (b) to understand how the integration in these profiles can help to explain specific workers’ outcomes (i.e., their perception regarding the human resources practice system applied and their commitment). However, whether using LPA or another approach, it is difficult to specify a priori what profile groups will emerge from a given sample [37]. Like Meyer and colleagues [37], we therefore base our hypotheses on the findings of earlier research. A summary of the profile groups obtained in the most recent studies is in Table 1.

In general, and regardless of the field of study (i.e., psychology applied to sports, education, or job relations), studies inspired by the SDT have typically identified three to five profile groups, and they either use scales that measure only the extrinsic and intrinsic motivation of the respondents, treating both motivations as two orthogonal constructs [33,38]; or, use scales that measure the motivation as a continuum that might go from amotivation to intrinsic motivation, considering external, introjected, identified, and integrated motivations [31,32,39].

Among the most commonly identified groups are a profile with high extrinsic and low intrinsic motivation, one with low extrinsic and low intrinsic motivation, one with high extrinsic and high intrinsic motivation, and another with low extrinsic and high intrinsic motivation. Some studies also identify a moderate or medium motivation profile [31,39], which represents individuals that scored moderately on all types of motivation (i.e., extrinsic, introjected, identified, integrated, intrinsic, and amotivation—when the last was also considered). Furthermore, these studies’ general conclusions are that individuals with more intrinsic profiles have more positive outcomes.

In the field of work and labor relations, using a measure with five motivation variables (i.e., external, introjected, identified, integrated, and intrinsic motivation), Moran and colleagues [39] arrived at a five-profile solution that presented differences regarding the individuals’ needs for satisfaction, job performance, and work environment perceptions. Their results included a low introjection profile (i.e., moderate scores on all types of motivation except in the introjected one), a moderately motivated profile (i.e., scored moderately on all types of motivation), a low autonomy profile (i.e., a particularly low means on integrated and intrinsic motivations), a self-determined profile (i.e., high intrinsic and low external motivation), and a motivated profile (i.e., high on all types of motivation). Findings showed that the self-determined cluster and the motivated cluster had more favorable results than the low autonomy cluster, which presented the worst scores. With regard to temporary work, De Jong and colleagues’ [9] analysis reached a three-profile solution that combined (a) voluntary motives or intrinsic choices (i.e., the freedom offered by temporary work); (b) involuntary motives or extrinsic choices (i.e., the difficulty of finding a permanent job), and (c) stepping-stone motives, which result from a combination of intrinsic and extrinsic choices (i.e., temporary work as a means to gain a permanent job). Their findings included an involuntary profile (high levels of involuntary and stepping-stone motives), a non-involuntary profile (all motives were rated as unimportant), and a stepping-stone profile (high scores of the stepping-stone and no reference to the voluntary or involuntary motives). More recently, [42] built a motivation profile typology of outsourcer and TAW in the contact center. Six profiles were identified: (a) involuntary motivation profile (i.e., high extrinsic and low intrinsic motivation; (b) moderate involuntary motivation (i.e., high extrinsic and identified motivation and medium integrated motivation); (c) low involuntary motivation (i.e., the extrinsic motivation was lower and closer to the integrated and identified motivation); (d) moderate voluntary motivation (i.e., high integrated and extrinsic motivation); (e) Voluntary Motivation (i.e., whit medium intrinsic and high extrinsic motivation); and (f) high motivation (i.e., a profile with high extrinsic and high autonomous motivations).We therefore anticipate finding the following profile configurations: (a) a profile with high extrinsic and low intrinsic, integrated, and identified motivations, which can indicate TAWs for whom this job is purely extrinsic; (b) a profile with low extrinsic and low intrinsic, integrated, and identified motivations, which may represent TAWs for whom their job is/was the only option available (i.e., a worker having no motivation at all); (c) a profile with high extrinsic and high intrinsic, integrated, and identified motivations, for workers who consider that their TAW job encompasses their needs and at the same time feel that it might be the only opportunity available to them; (d) a profile with low extrinsic and high intrinsic, integrated, and identified motivations that may include TAWs who see their work arrangement as an advantage (i.e., the job is almost a personal choice); and (e) a profile with moderate or medium motivation scores on all types of motivation that may include TAWs who are taking some advantage from the work situation.

### 1.2. Different Motivations, Different Employment Relationships

Earlier research has gathered evidence that TAWs value the investments made by the organizations, especially when those investments have an impact on what concerns them the most [43]. For instance, when TAWs perceive that the HRP system has a concern for their needs and goals, they tend to reciprocate in the same way [44]. Earlier studies report how investments made in TAWs can especially contribute to an increase in their affective commitment [45,46]. Moreover, in the specific case of TAWs, they establish a double employment relationship that involves two different entities [47] and, therefore, they develop two different affective commitments, each in response to the support they receive from each organization—the agency and the client-company [15,18]. When TAWs have a favorable perception of the support they receive from the agency, they increase their commitment toward it [15,18]; and the same happens with the client company [15,18,48]. However, the construction of workers’ perceptions is not linear and it can depend on their motivation to work, or more importantly, on the combination of their different motivations to work.

It is believed that the voluntariness of temporary workers is directly linked with their professional outcomes. In fact, individuals who voluntarily choose to become temporary workers reveal higher levels of work satisfaction when compared with those whose choices were involuntary [49,50]. As pointed out by [51], the underlying reason for taking a temporary work assignment may indeed influence workers’ outcomes. According to this research, temporary workers who desire to gain experience or learn useful skills showed more positive organizational citizenship behavior than those whose only goal was to earn money [51].

As mentioned above, with regard to the motivation profiles research, studies have also shown that when individuals feel that their motivation is somehow less external and more intrinsic, or incorporated as such, they behave in a more positive way toward their context [31,32,33], even when talking about work relations [38,39]. A motivation profile analyses of Moran and colleagues [39] and Van den Broeck and colleagues [38] corroborate the existence of different work attitudes among different profiles. According to their findings, profiles with high intrinsic motivations had more favorable results than those with high extrinsic motivations. In the end, intrinsic motivations seem to have a buffering effect on the extrinsic motivation: the negative effect of this motivation diminishes in the presence of more intrinsic ones. Concerning TAWs’ motivation, De Jong and colleagues [9] also showed that different profiles relate differently with work involvement and employability perception. The non-involuntary profile (i.e., workers rating all motives (involuntary, stepping-stone, and voluntary) as unimportant) was the most distinct one: respondents in this cluster presented low levels of work involvement (although the difference was not significant) and higher levels of employability. According to the authors, respondents in this profile could be classified as passive or ambivalent toward their work situation, representing, what they called, a “resigned group” (i.e., works with long tenure and relatively low work involvement and may feel that they have no other work options, and as a result, they cannot identify any motivation to be there) [9]. Finally, the results of Sobral and colleagues [42] show that the motivation profile to which TAW and outsourcing workers belonged to, were able to differentiate their perceptions over the HRP and their affective commitment. In general, both outsourcing workers and TAW profiles with a higher presence of intrinsic motivations presented better outcomes.

Transposing these findings to the analysis of TAWs’ motivation profiles, we consider that TAWs in different motivation profiles will present different perceptions of the HRP system and different levels of affective commitment. Moreover, we assume that when TAWs have only an extrinsic motivation, they have a poorer employment relationship; but when they have an extrinsic motivation combined with more intrinsic motivations—intrinsic, integrated, identified—they can show a more positive employment relationship.

The typology of TAWs’ motivation is related to their perception of HRP, and an affective commitment with the agency and the client company.

**Hypothesis** **1.**
*Five profile motivation groups having distinct patterns of intrinsic, integrated, identified, and extrinsic motivations exist within the TAWs sample, showing different levels of motivation.*


**Hypothesis** **2.**
*TAWs in profiles with a higher presence of intrinsic motivations (i.e., intrinsic, integrated, and identified motivations) have a higher perception of HRP and greater affective commitment toward the agency and client company—than those in a more extrinsic profile (i.e., profiles with a lower presence of intrinsic motivations).*


## 2. Method

### 2.1. Sample

The sample comprises TAWs (N = 3766) from seven temporary employment agencies operating in Portugal. The data were collected online with the use of a commercial survey service—survey monkey. An e-mail containing the link to the survey was addressed by the agencies to all their workers. In this e-mail the future participant of the study found a message from the research team in which all the procedures were explained. Respondents were assured that their answers were confidential and anonymous. Participants were also informed that they would have the opportunity to receive the overall results. These instructions were written on the questionnaire’s cover letter. The instructions explained that the questions were directly related to several parts of their daily work, specifically their perceptions of employment relationships. Participants were informed that the questionnaire was not a test and that there were no right or wrong answers. Workers were also assured that the companies would only have access to a final report and not to the raw data, as the data were used exclusively for academic research. The lead researcher’s email address was included in the cover letter in addition to a website address where respondents could find more information about the research project, including the involved academic organizations, its goals, outcomes, partners and other researchers included in the process. There was no incentive (cash or otherwise) for participating in this project. Because participation was voluntary and anonymous, participants did not sign an informed consent form. The participation rate varied between 42% and 58% among the various companies.

The sample contains workers from several sectors (26.1% blue collar workers; 56.1% white collar workers; 17.8% not identified) with different functions and backgrounds (50.9% previously unemployed workers; 13.4% workers from previous temporary agency; 16.9% previously hired with a non-temporary contract; 6.4% previously self-employed workers; and 12.4% in a first job experience). In common, they all work in client companies at which the decision to employ TAWs was based on the need to adapt to current market needs. This decision enabled the organizations to adjust to fluctuations in client requests or services rendered.

Regarding the demographic characteristics, women are 55.6% of the sample and the average age is around 31 years old. The distribution by academic qualifications is: up to 9th grade 9.8%; secondary 23.7%; university attended 29.1%; graduated 18.4%; postgraduate studies 19%. The distribution by tenure in the agency is: less than 3 months 21.7%; 3 to 6 months 19.2%; 6 to 9 months 12.9%; 9 months to a year 10.9%; 1 to 5 years 31.4%; 5 to 10 years 2.9%; more than 10 years 0.9%. The distribution by tenure in the client company is: less than 3 months 21.5%; 3 to 6 months 18.1%; 6 to 9 months 12.6%; 9 to 13 months 10.6%; 13 to 18 months 9.1%; more than 18 months 28.2%. The respondents completed an online questionnaire in which the anonymity of their responses was assured.

### 2.2. Measures

TAWs’ motivation. We measured motivation using a version of the MAWS, originally developed by Gagné and colleagues [52] and duly translated, adapted, and tested with TAWs by Lopes and Chambel [25]. The measure makes it possible to distinguish between four types of motivation: intrinsic (4 items; “Because I like to be temporary”; α = 0.88); integrated (4 items; “Because it is the job that best adapts to my needs”; α = 0.81); identified (4 items; “I choose to be temporary because it can allow me to get a permanent job”; α = 0.81); and extrinsic (4 items; “Because it allows me to gain more money”; α = 0.80). The items were answered on a 7-point Likert rating scale that ranged from “Not at all” (1) to “Completely” (7).

Affective commitment. We measured TAWs’ affective commitment toward the agency (α = 0.84) and the client company (α = 0.90) using a version of the scale built by Meyer, Allen, and Smith [53]. The 6-item scale includes, for each commitment, sentences like: “I would be happy if I developed the rest of my career in this company” or “I do not feel emotionally attached to this company” (inverted). The items were answered on a 7-point Likert scale that ranged from “I totally disagree” (1) to “I totally agree” (7). This scale had already been translated, adapted, and tested with Portuguese TAWs [45].

Perception of HRP. Perceptions of the HRP system was measured using a 20-item scale (α = 0.87) that measured TAW’s socialization, recruitment, training (i.e., promotion of internal and external employability), and performance appraisal [54,55]. This scale was adapted, tested and validated with TAW by Sobral and colleagues [56], and was previously tested with Portuguese TAW [44]. According to the authors [56], because each HRP is not independent, this scale examines how TAW’s perceive HRP regardless of the company providing the practices (i.e., agency or client company). It is believed that the agency and the client are jointly responsible for the HRP system; thus, TAW’s overall perceptions were measured. Example items of the HRP included: “When I started working in this company, my job goals were clearly explained to me”, “The criteria for performance evaluation are clear at this company”, and “With the experience/training that I have received, I can easily change jobs inside the company where I am now”. The items were answered on a 7-point Likert scale that ranged from “I totally disagree” (1) to “I totally agree” (7).

### 2.3. Analyses

Our data analysis involved five phases. First, we performed two confirmatory factor analyses (CFA) to examine the factor structure of the items composing the affective commitment and the perception of HRP. These analyses where conducted with AMOS version 21.0 (Amos, Chicago, IL, USA).

Second, we performed a CFA to evaluate the discriminant validity of all the self-report measures and addressed concerns about common method bias by comparing our model with a one-factor model [57].

Third, we use the statistical software MPlus v.7 to conduct the LPA and identify the latent TAW motivation profiles. LPA decomposes the covariances to highlight the relationships among individuals instead of variables [58] and involves objective criteria in decisions regarding the number of subgroups extracted [59]. Following Nylund and colleagues [60], the optimal profile number was determined through an iterative process in which a two-profile model was estimated and successive profiles were then added. An optimal profile number must exhibit the following characteristics: (a) the lowest BIC; (b) a significant LMR p-value; (c) clearly defined profiles, with high probability of endorsement; (d) an entropy value close to 1; (e) theoretical agreement; and (f) no profiles with a small number of individuals.

Fourth, we conducted ANOVA analyses to evaluate the uniqueness of the motivation generated by the LPA. In the last phase we conducted one-way ANCOVAs to test the possible differences among profiles regarding the workers’ commitment, the wellbeing at work, and the perception of HRP. Like ANOVA, ANCOVA is a statistical model, but one that allows us to use a general linear model and to compare a variable in various groups taking into account the variability of other variables, called covariances.

## 3. Results

### 3.1. Confirmatory Factor Analysis

Regarding the affective commitment items, the overall goodness of the fit was based on combinations of several fit indices. The model had an adequate fit to the data when there was a significant chi-square, 0.90 or higher for the Tucker Lewis (TLI) and fit indices (CFI), 0.06 or less for the root mean square error of approximation (RMSEA), and 0.08 or less for the standardized root mean square (SRMR). The model with one latent factor had a good fit for the affective commitment with the agency (χ^2^ (7) = 182.00, *p* < 0.001; SRMR = 0.07; CFI = 0.99; TLI = 0.97; RMSEA = 0.08), and with the client company (χ^2^ (6) = 185.48, *p* < 0.001; SRMR = 0.06; CFI = 0.99; TLI = 0.97; RMSEA = 0.09). As to the HRP items, the model with one latent factor (i.e., HRP system) had a good fit (χ^2^ (140) = 2134.57, *p* < 0.001; SRMR = 0.06; CFI = 0.96; TLI = 0.95; RMSEA = 0.06). The resulting 20-item scale had a reliability of 0.87, which is comparable to the one that Takeuchi and colleagues [52] obtained for their HR system scale (0.90).

The CFA conducted to evaluate discriminant validity and test for common method variance among the self-report measures revealed an acceptable fit for our four-factor theoretical model (χ^2^ (1008) = 9538,73, *p* > 0.001; SRMR = 0.06; CFI = 0.93; TLI = 0.92; RMSEA = 0.05). We compared this model with the one-factor model in which all items were loaded on a single latent variable [χ^2^ (1019) = 26,173.04, *p* < 0.001, SRMR = 0.11; CFI = 0.79, TLI = 0.77; RMSEA = 0.08]. Our theoretical model provided a better fit to the data [∆χ^2^ (11) = 16,634.31, *p* < 0.001], indicating that the majority of variance in data cannot be explained by a single factor. We further tested an additional model (methods model) in which an unmeasured latent methods factor was added to the four-factor theoretical model. In this model, all items were loaded on their theoretical constructs, as well as on the latent methods factor. The methods model obtained a good fit (χ^2^ (970) = 7109.73, *p* < 0.001; SRMR = 0.07; CFI = 0.95; TLI = 0.94; RMSEA = 0.04), and the method factor accounts for 13.6% of the variance, which falls below the threshold of 50% [61]. Although both models include the same observed variables, the methods model cannot be nested within the one-factor model, and for that reason we calculated the CFI difference to compare the goodness-of-fit of these models. The change of CFI between both models was 0.02, which is below the suggested rule of thumb of 0.05 [62]. Therefore, we conclude that common method bias is not a major concern in this study. Means, standard deviations, and correlations among the study variables are shown in Table 2, and the presence of distinct constructs can be observed.

### 3.2. Identification of TAWs’ Motivation Profiles

The optimal model included five profiles (Figure 1) and presented: (a) low BIC (44,858.226) and LMR (−22,313.840) values; (b) a significant LMR *p*-value; (c) clearly defined profiles—individuals*’* probabilities of belonging to the assigned profiles were high (0.84 to 0.95) while the probabilities of belonging to other groups were low (0.00 to 0.12); (d) an entropy value close to 1 (0.85); (e) a pattern of means consistent with earlier findings that is theoretically meaningful; and (f) a sufficient number of cases in each profile group to warrant retention (range = 136 to 2941).

The results of the ANOVAs conducted to compare motivation levels across profile groups revealed significant differences between intrinsic motivation (F = 3275.69, *p* < 0.001; η^2^ = 0.78), integrated motivation (F = 1479.12, *p* < 0.001; η^2^ = 0.78), identified motivation (F = 496.95, *p* < 0.001; η^2^ = 0.78), and extrinsic motivation (F = 416.44, *p* < 0.001; η^2^ = 0.78). The distribution of the motivation means across profiles fulfilled the normality assumption, with all the skewness values ranging between −3 and 3, and all the kurtosis values ranging between −10 and 10 [63].

The name assigned to each motivation profile was decided through the results of the ANOVAs and post-hoc comparisons (Table 2).

In the end, reaching a five-profile model is consistent with our first hypothesis, which points toward a TAW motivational profile having three to five profile groups. However, contrary to our hypothesis, the results show a predominance of the extrinsic motivation in most of the groups (Table 2). Nonetheless, it is possible to discern and statistically establish several significant variations between the profiles regarding each type of motivation. In other words, we were able to find significantly distinct patterns of intrinsic, integrated, identified, and extrinsic motivations inside our sample. Each of the five profiles combines the motivations in a significantly different way. Moreover, findings show that the probability of each individual to be included in one particular profile is significantly higher than that of being included in any other.

The Involuntary Motivation Profile (Profile 1) presents the greatest variation between the extrinsic motivation and all the other motivation variables (i.e., high extrinsic and low intrinsic, integrated, and identified motivations). The Low Motivation Profile (Profile 2) scores low in all the motivations (i.e., TAWs belonging to this group do not have a specific reason to make this choice). The Moderate Involuntary Motivation Profile (Profile 3) emphasizes (as do most of profiles except the Low Motivation Profile) the extrinsic motivation choice (nonetheless, the difference between extrinsic, identified, and integrated motivation is much lower here than the one in the Involuntary Motivation Profile). For TAWs in the Moderate Involuntary Motivation Profile, their work arrangement occurs not only because they have no other choice, but also because they see it as a chance: (a) to adjust labor with their personal needs (e.g., study and family); and (b) to try to obtain a permanent job. The Moderate Motivation Profile (Profile 4), shows that it is possible to find a fairly positive scoring (above 3.5) within the integrated, identified, and extrinsic motivations alongside a score above 3 for the intrinsic motivation. Finally, in the High Motivation Profile (Profile 5) intrinsic, integrated, identified, and external motivations all score above 4.

When comparing the anticipated profile configuration with the profile configuration found, it is clear that it was not possible to find (a) a profile with low extrinsic and high intrinsic, integrated, and identified motivations; and (b) a profile with moderate or medium motivation scores on all types of motivation. Even so, the Moderate Motivation Profile presents scores above the scale mid-point for integrated, identified, and extrinsic motivations. Thus, H1 is partially supported.

### 3.3. Analyses of Covariance

The correlations in Table 2 justify the use of gender, age, qualifications, seniority in the agency and in the client-company, and the agency in which TAWs work as control variables in the test of H2. Therefore, we chose to use a one-way ANCOVA to test the relationship of each profile (independent variable) with TAWs*’* affective commitment toward the agency and the client company, and TAWs*’* perception of HRP (dependent variables). Before doing so, we confirmed that the distribution of the motivation means across profiles fulfilled the normality assumption, with all the skewness values ranging between −3 and 3, and all the kurtosis values ranging between −10 and 10 [63]. As postulated in H2, the one-way ANCOVA revealed significant differences between the profiles with regard to affective commitment toward the agency (F = 102.81, MSE = 1.91, *p* < 0.001; η^2^ = 0.11), affective commitment toward the client company (F = 54.69, MSE = 2.02, *p* < 0.001; η^2^ = 0.07), and perception of HRP (F = 98.26 MSE = 1.17, *p* < 0.001; η^2^ = 0.11). Given that all the variables showed the overall effects, we conducted post-hoc pair-wise comparisons (Table 3).

### 3.4. Differences between TAWs’ Motivational Profiles

In accordance with H2, profile groups with more intrinsic motivations (i.e., profiles with a higher presence of intrinsic, integrated, and identified motivations) present a more positive employment relationship than those with extrinsic motivation. Consequently, the High Motivation profile achieved the best results on all variables, followed by Moderate Involuntary Motivation and Moderate Motivation—which have a high presence of integrated and identified motivation. The Moderate Involuntary Motivation and the Moderate Motivation profiles lie in an intermediate position with respect to TAWs*’* employment relationship, showing that TAWs with positive levels of integrated and identified motivations show a positive employment relationship. Moreover, in accordance with our hypotheses, profiles that join intrinsic and extrinsic motivations present more positive employment relationships than those that present only an extrinsic motivation. Therefore, despite the prevalence of extrinsic motivations among TAWs, our results also show that when the extrinsic motivation is accompanied by more intrinsic motivations (i.e., integrated and identified) it produces a buffering effect on the TAWs*’* employment relationship. After the High Motivation Profile, Moderate Involuntary Motivation and the Moderate Motivation profiles present the best scores.

In opposition, Involuntary Motivation and Low Motivation profiles present the poorest results. TAWs that present an involuntary motivation or a low motivation are those who have more difficulties in managing their employment relationship and in achieving a broadly positive perception of their job.

Looking specifically at affective commitment scores, it is worth noting that all profiles score higher in the commitment toward the client company than toward the agency. Moreover, and with the exception of the High Motivation profile (which always presents high levels for all motivations), the profile with the highest level of affective commitment toward the client company is the Moderate Involuntary Motivation, which has the highest level of identified motivation and the third highest level of integrated motivation. This effect points out that TAWs with high integrated and identified motivation may see their current job as an opportunity to move to a permanent position inside the client company (stepping-stone effect).

Finally, the perception of HRP results for each profile shows an average outcome above 5 (from 1 to 7) for the Moderate Involuntary Motivation, Moderate Motivation, and High Motivation profiles (Table 2), indicating that TAWs with more integrated and identified motivations are those who more positively perceive the applied practices.

## 4. Discussion

The key goal in the present study is to create a TAW motivational profile that identifies several different levels of motivation in accordance with self-determination theory [11], and analyze how each profile associates its members’ employment relationship (i.e., affective commitment toward the agency and the client company and perception of HRP).

In response to our first hypothesis, which implied the establishment of different motivation profiles inside the total sample, as expected it was possible to significantly distinguish five groups. However, the five profile groups identified in the present sample do not present the same characteristics as those found in earlier profile studies [31,32,33,39]. We did not find a profile with low extrinsic and high intrinsic motivations, nor a profile with moderate scores on all types of motivation. In line with De Jong and Shalk [64], and Sobral and colleagues [42] who reported the prevalence of external motivation in TAW, our results show a high presence of extrinsic motivation in the majority of profiles (with the exception of the profile scoring low in all the motivations, i.e., the Low Motivation Profile).

More important, though, was the existence of intrinsic, integrated, identified, and extrinsic motivations inside the same profile, which reinforced the belief that different types of motivation are not necessarily exclusive and can co-exist with each other [32,42].

In the end, we found a profile with a high extrinsic and low intrinsic, integrated, and identified motivations (i.e., Involuntary Motivation profile; TAWs with the lowest ability to incorporate external regulation), a profile with low extrinsic and low intrinsic, integrated, and identified motivations (i.e., Low Motivation profile; TAWs who do not present any motivation to be in their current work situation), and a profile with high extrinsic and high intrinsic, integrated, and identified motivations (i.e., High Motivation; TAWs who voluntarily wish to be in their current work situation but who also incorporate the external reasons as their own). We also found a profile with Medium Motivation in which integrated, identified, and extrinsic motivations score above the scale mid-point, and the intrinsic motivation achieves a score slightly below the scale mid-point; and a profile with a high extrinsic, integrated, and identified motivation (i.e., Moderate Involuntary Motivation). In these last two profiles we found high levels of integrated and identified motivation, showing that they include TAWs who look at their job as a chance to gain some tools, experience, and/or connections to reach what they see as a better situation and possibly a permanent position.

The presentation of five profiles through which TAWs can expose their motivations is one of the most significant contributions of this research. Indeed, this is possible due to (a) the definition of a very concrete target and goal, presenting a profile typology that could report TAWs’ work motivations; and (b) the use of a motivation measure built and tested specifically for TAWs [25] and with a wider range of responses. By comparing these results with those obtained by De Jong and colleagues [9], it seems clear that it was possible to overcome the small dimension of their measure and the imprecision of their typology proposal, (deriving from the use of a complex sample containing very different proportions of contingent workers with different types of contract).

This research also contributes to the discussion about the implication of SDT on the TAWs’ motivations, and on their employment relationship construction and perception. In agreement with earlier studies [31,32,33,65], our findings suggest that it is possible to study various combinations of intrinsic and extrinsic motivations, since workers can simultaneously display different types of motivation or behavioral regulation [66]. We found subgroups of TAWs for whom these types of motivation appeared to co-exist quite readily. According to González and team [32], this kind of result shows that different types of motivation can be “mutually compatible rather than inherently exclusive”.

As in the outcomes and construction of work motivation profiles of Moran and colleagues [39], Van den Broeck and colleagues [38], and Sobral and colleagues [42], we also found a buffering effect, meaning that the presence of intrinsic motivations can attenuate the negative effect of the extrinsic motivations on the individual’s self-determination. When TAWs are able to integrate the reasons for being in this specific work arrangement, the negative effect of the extrinsic motivation can be decreased by the presence of more intrinsic motivations. Our findings suggest that being high in integrated and/or identified motivation, along with being high in the extrinsic motivation, is associated with a more favorable employment relationship. If a worker identifies himself/herself with his/her work and/or really enjoys performing it, the motivation coming from the earnings may be secondary [39].

Profiles with high levels of integrated and identified motivation along with high levels of extrinsic motivation (i.e., Moderate Involuntary Motivation and Moderate Motivation profiles) or with high levels of intrinsic, integrated, and identified motivations along with high levels of extrinsic motivation (i.e., High Motivation profile) respond with higher levels of affective commitment and a better perception of the HRP, showing the existence of a buffering effect of the intrinsic motivations over the extrinsic ones. Profiles with low levels of intrinsic motivation along with high levels of extrinsic motivation (i.e., Involuntary Motivation profile) or with low levels of the intrinsic motivations along with low levels of extrinsic motivation (i.e., Low Motivation profile) present a poorer employment relationship regarding the levels of affective commitment and perception of the HRP.

In fact, the use of the LPA allowed us to reveal unobserved heterogeneity in this specific population, finding groups with similarity in their responses [34]. Through this method we achieved more complex analyses and reached more enlightening conclusions than the studies that have focused only on the dichotomy between voluntary and involuntary reasons to be in a temporary work arrangement [14]—including our detection of moderate profiles with a strong presence of identified and integrated motivations or reaching a profile in which all the motivations have a strong presence. Thus, and although the main reasons for being a TAW are related to extrinsic or involuntary motives, we see that it is possible for these workers to develop a positive employment relationship. TAWs are able to incorporate the external regulation and accept a situation that is not what they really want, but they recognize the advantages of being employed and the opportunity to satisfy their aspirations to obtain a better/different job [9].

On the other hand, and reinforcing the idea of high involuntariness among TAWs, we see that regardless of the profile, TAWs always present a stronger commitment with the client company than with the agency, meaning that the connection between these workers and the company in which they provide a service is something that seems to acquire a great importance. Moreover, the high presence of the identified motivation in all profiles—even the Involuntary Motivation and the Low Motivation profiles present in some degree of identified motivation—show that either voluntarily or involuntarily, TAWs rely on temporary employment to obtain a permanent position [9] within the client company. In fact, it is known that the temporary arrangements are often used by the client company as a tool for recruitment and selection [67].

Overall, our findings reveal that a task can be performed for intrinsic and extrinsic reasons simultaneously [68], and studies on motivation should not rely on a unidimensional model that reflects only the degree to which individuals endorse intrinsic over extrinsic reasons to perform an activity [33]. Therefore, intrinsic and extrinsic motivations appear to be mostly independent, rather than opposite poles of a single dimension [65]. As suggested by González and colleagues [32], the effect that intrinsic motivations can have on the extrinsic may provide the framework for companies (agency and client companies) to design intervention programs in which TAWs can be encouraged, through an internalization process, to progress from extrinsic toward more intrinsic motivations. Indeed, studies demonstrate that it is possible to change suboptimal forms of motivation into more self-determined ones [69]. Small events that are repeated several times and lead to gradual and internalized behaviors allow people to experience more adaptive outcomes. Future research should use a longitudinal study to approach the issue of motivational change on TAWs.

## 5. Study Limitations

Concerning the data collection method (i.e., online questionnaires), and despite the considerable size of the sample (N = 3766), this study cannot be considered to be representative. First, individuals who responded to our questionnaire had to have Internet access and informatic knowledge, which may have restricted the sample to younger and scholarly respondents, in a less-peripheral job position. Second, the study focus is on Portuguese TAWs, and as result, extending and generalizing our findings to countries outside Europe should be done with considerable caution. But even if looking to these findings as a trend within the European Union, it is necessary to pay attention to the specificities of the Portuguese labor market. Chambel and Sobral [70] pointed out three aspects that should be considered when analyzing the results: (1) the last economic crisis (before COVID 19), caused high levels of unemployment and restricted recruitment activities that limited the chances of obtaining a permanent contract; (2) Portuguese workers value job security and prefer contexts where they can find job stability; and (3) since temporary agency work does not fulfill these values, the motives to choose this employment situation will be predominantly extrinsic. Even so, we believe that our study illustrates the importance of TAWs’ motivations in their employment relationship. Future studies should approach this issue using representative samples and should consider a cross-cultural study.

Furthermore, the use of cross-sectorial and self-reported data also influences our results. The cross-sectorial design limited the possibility to make causal inferences. Thus, future research may seek, through a longitudinal method, to empirically test the dynamic characteristic of motivation profiles. Furthermore, the use of self-reported variables may raise questions of common-method bias. To minimize this limitation we followed several methodological and statistical recommendations [57,71], such as: (a) ensuring the adequacy and clarity of the instrument by asking academic experts and HRM professionals (i.e., HR directors of temporary agencies and client companies) to review the questionnaires; and (b) assuring the respondents that their answers were confidential and that there were no right or wrong answers.

Lastly, and because the invitation to be part of the study came from the temporary agencies’ management, the responses could be influenced by the social desirability effect. To prevent this possible limitation, we guaranteed that the respondents were fully aware of the project’s goals and that it was totally conducted by an independent institution. As a consequence, the results show a significant variance of the responses regarding the different variables and dimensions that were measured.

## 6. Conclusions

In today’s globalized economy, companies look for human flexibilization as a way of obtaining a competitive advantage [3]. In fact, the increasing use of contingent work in the last few decades can be considered as more than an economic matter; it can be considered as a reflex of on-going social change regarding contingent contracts [8]. The introduction and growth of labor market flexibility has brought new challenges to HR management.

Studies have proven that it is in the best interest of the organizations to ensure that all workers are well treated because only in that way is it possible to obtain positive responses and outcomes [44,46]. In the specific topic of motivations, this study shows that TAWs are able to simultaneously point to intrinsic and extrinsic reasons to be in their work arrangement. More importantly, if they are able to value their job and internalize the extrinsic reasons to be in it, through the growth of identified and integrated motivations, they present a better employment relationship. Being able to motivate workers, regardless of their contract type, seems to contribute to the organization’s efficiency since it is related to workers’ commitment and perception of HRP.

## Figures and Tables

**Figure 1 ijerph-18-06779-f001:**
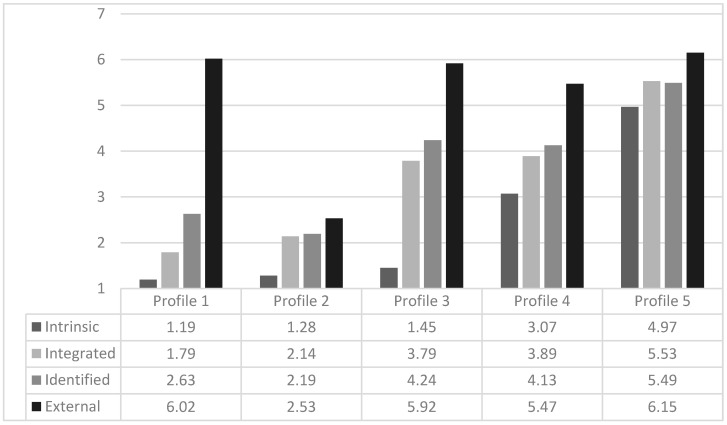
Characteristics of the latent profiles on the indicators.

**Table 1 ijerph-18-06779-t001:** Summary profiles obtained in earlier research.

	HC/LA	LI	C&A Low	M/M	C&A High	HA/LC	M	Method
Ntoumanis [40]	✓			✓		✓		Cluster analysis
Hayenga and Corpus [33]	✓		✓		✓	✓		Cluster analysis
Mouratidis and Michou [41]	✓		✓		✓			Cluster analysis
González et al. [32]	✓		✓		✓	✓		Cluster analysis
Gillet et al. [31]			✓	✓	✓			Cluster analysis
Moran et al. [39]	✓	✓		✓		✓	✓	Cluster analysis
Van den Broeck et al. [38]	✓				✓	✓		Cluster analysis
	NonInv	HInv/HSS	HSS					
De Jong et al. [9] (TAWs)	✓	✓	✓					LPA

Note: HC = high control motivation; LA = low autonomy motivation; C&A = control motivation and autonomy motivation; HA = high autonomy motivation; LC = low control motivation; LI = low introjection motivation; M/M = medium motivation; M = motivation; NonInv = Non-involuntary motivations; HInv = high involuntary motivation; HSS = high stepping-stone motivation.

**Table 2 ijerph-18-06779-t002:** Descriptive statistics and correlations.

Variables	Mean	SD	1.	2.	3.	4.	5.	6.	7.	8.	9.	10.	11.	12.	13.	14.	15.	16.	17.	18.	19.
1. Intrinsic	1.79	1.13																			
2. Integrated	2.83	1.46	0.63 **																		
3. Identified	3.40	1.52	0.45 **	0.53 **																	
4. Extrinsic	5.77	1.20	−0.06 **	0.01	0.15 **																
5.Agency Com.	3.94	1.38	0.33 **	0.36 **	0.43 **	0.06 **															
6. Client Com.	4.43	1.49	0.19 **	0.26 **	0.45 **	0.11 **	0.55 **														
7. HRP	4.93	1.16	0.25 **	0.37 **	0.51 **	0.13 **	0.46 **	0.52 **													
8. Age	30.7	8.13	0.04 **	−0.06 **	0.03	0.04 *	0.10 **	0.09 **	−0.01 *												
9. Gender ^a^	0.44	0.50	0.01	0.00	0.03	−0.05 **	0.00	−0.03	−0.01 **	0.01 **											
10. Qualif.	3.18	1.33	−0.17 **	−0.14 **	−0.13 **	−0.11 **	−0.24 **	−0.11 **	−0.09 **	−0.12 **	−0.18 **										
11. Sen. ag.	3.24	1.67	−0.06 **	−0.05 **	−0.13 **	0.07 **	−0.05 **	0.00	−0.06 **	0.01 **	−0.06 **	−0.05 **									
12. Sen. client	3.52	1.94	−0.08 **	−0.05 **	−0.13 **	−0.07 **	−0.07 **	0.04 **	−0.05 **	0.09 **	−0.06 **	−0.02	0.79 **								
13.Agency 1	*n* = 451	---	−0.03	0.01	0.01	−0.01	−0.05 **	0.01	−0.04 *	−0.03 *	0.03	−0.02	−0.12 **	−0.12 **							
14. Agency 2	*n* = 539	---	0.02	0.04 *	0.06 **	−0.01	0.03 *	0.01	0.07 **	−0.01	0.01	0.09 **	−0.16 **	−0.16 **	−0.15 **						
15. Agency 3.	*n* = 207	---	−0.07 **	−0.05 **	−0.04 *	0.00	−0.04 *	−0.02	−0.05 **	0.05 **	−0.02	0.14 **	0.02	0.01 **	−0.09 **	−0.01 **					
16. Agency 4	*n* = 332	---	0.02	0.03	0.04 *	−0.03	0.04 **	0.04 *	0.01	0.03 *	−0.01	0.03	0.02	0.01	−0.12 **	−0.13 **	−0.08 **				
17. Agency 5	*n* = 978	---	−0.04 *	−0.03 *	−0.02	0.04 **	−0.01	0.00	−0.00	−0.08 **	−0.09 **	−0.11 **	0.17 **	0.13 **	−0.30 **	−0.34 **	−0.11 **	−0.26 **			
18. Agency 6	*n* = 588	---	0.025	−0.003	−0.066 **	−0.016	−0.014	−0.042 *	−0.016	0.036 *	0.065 **	0.043 *	0.049 **	0.066 **	−0.159 **	−0.176 **	−0.104 **	−0.13 **	−0.35 **		
19. Agency 7	*n* = 535	---	−0.020	−0.002	−0.024	0.012	−0.005	0.004	0.017	−0.020	−0.053 **	−0.095 **	0.154 **	0.142 **	−0.150 **	−0.166 **	−0.098 **	−0.13 **	0.497 **	−0.18 **	
20. Other	*n* = 136	---	0.114 **	0.027	0.045 **	−0.015	0.058 **	0.006	0.012	0.094 **	0.095 **	−0.141 **	−0.092 **	−0.096 **	−0.071 **	−0.079 **	−0.047 **	−0.06 **	−0.16 **	−0.08 **	−0.08 **

^a^ 0 if Female and 1 if Male; * *p* < 0.05; ** *p* < 0.01; Note: *n* = 3766; Qualif. = Qualifications; Sen. Ag. = Seniority Agency; Sen. client = Seniority client; Other = unidentified agencies.

**Table 3 ijerph-18-06779-t003:** Motivation and work variable means associated with the 5-profile model.

	Involuntary Motivation	Low Motivation	Moderate Involuntary Motivation	Moderate Motivation	High Motivation	Post-Hoc Comparisons
n	1877	136	831	755	167	
%	49.8	3.6	22.1	20.0	4.4	
Intrinsic	1.19	1.28	1.45	3.07	4.97	5 > 4 > 3 > 2, 1
Integrated	1.79	2.14	3.79	3.89	5.53	5 > 4 > 3 > 2 > 1
Identified	2.63	2.19	4.24	4.13	5.49	5 > 3, 4 > 1 > 2
Extrinsic	6.02	2.53	5.92	5.47	6.15	5 > 1 > 3 > 4 > 2
Agency Commitment	3.47	3.41	4.35	4.54	5.29	5 > 4, 3 > 1, 2
Client Commitment	4.13	3.94	4.89	4.75	5.40	5 > 4,3 > 1, 2
HPR	4.57	4.23	5.34	5.20	5.81	5 > 3 > 4 > 1 > 2

Post-hoc comparisons indicate which means differ significantly at least *p* < 0.05; Note: 1 = Involuntary Profile; 2 = Low Motivation Profile; 3 = Moderate Involuntary Profile; 4 = Moderate Motivation Profile; 5 = High Motivation Profile.

## Data Availability

The data presented in this study are available on request from the corresponding author.

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
