# Peer review of "The Temporary Agency Worker’s Motivation Profile Analysis"

_ijerph, 2021, doi:10.3390/ijerph18136779_

Round 1
Reviewer 1 Report
I think it is an excellent article.
Below are some recommendations for improvement:
The headings, such as the one on line 111, should perhaps be numbered, as is the title of the introduction on line 27. This aspect should be made uniform.
“Possible to”, separated words, in line 160.
The two hypotheses (233-235 and 301-304) can be formulated in sequence, one after the other, since the literature review (236-300) is valid for both.
“Temporary workers volition” (255-) seems to be a key aspect. However, it has not been measured in the study. The decision not to measure it should be justified. Another option may be to not develop this aspect so much in the literature review.
The proportion of participants in the invited sample is surprisingly high (314-315). Perhaps it would be useful to include some reflection on the motivation of the subjects to participate in the study, especially considering the possible effect of the social desirability of the responses, bearing in mind that the invitation came from the company's management. It can be argued that the responses have a significant variance in the different dimensions.
It would be good if they could provide information on the fulfillment of the normality assumptions. If there were difficulties in this regard they could use boostrapping techniques. All this is not essential, only recommended.
In line 357-358 two symbols are missing, I suppose chi2 and p-value, idem in 376 and 414.
It is convenient to use 3 decimal places in places such as lines 358-359.
The CFAs are reported under two different sections, it is a bit confusing for the reader.
“in line with our predictions” would bring this type of commentary into the discussion, I would avoid it in the results section.
Line 491: would use the diagonal of the table to incorporate these values.
In the literature review and/or discussion I would perhaps add some reference to the different economic and labor regimes and work culture in which the reported studies were carried out, especially about the specificity of the Portuguese labor market of this particular study. Just if posible.
It is very important to report the effect size values, eta squared for analysis of variance and Cohen's d for t-test.
The use of t-test is announced (395) but I cannot find it in the results.
Author Response
Reviewer’s 1 comments and questions:
Comments to the Author
- 1. I think it is an excellent article
Authors’ answer:
We are pleased with the Reviewer’s overall positive evaluation of the paper. Thank you very much!
Specific items to consider:
2. The headings, such as the one on line 111, should perhaps be numbered, as is the title of the introduction on line 27. This aspect should be made uniform.
Authors’ answer:
The headings are now numbered in accordance with the Reviewer’s 1 consideration.
- "Possible to", separated words, in line 160.
Authors’ answer:
This spelling error is already corrected.
- "The two hypotheses (233-235 and 301-304) can be formulated in sequence, one after the other, since the literature review (236-300) is valid for both.”
Authors’ answer:
The two hypotheses are now formulated in sequence, one after the other:
“Hypothesis 1. Five profile motivation groups having distinct patterns of intrinsic, inte-grated, identified, and extrinsic motivations exist within the TAWs sample, showing dif-ferent levels of motivation.
Hypothesis 2: TAWs in profiles with a higher presence of intrinsic motivations (i.e.: in-trinsic, integrated, and identified motivations) have a higher perception of HRP and greater affective commitment toward the agency and client company – than those in a more extrinsic profile (i.e. profiles with a lower presence of intrinsic motivations).”
- 5. “Temporary workers “volition" (255-) seems to be a key aspect. However, it has not been measured in the study. The decision not to measure it should be justified. Another option may be to not develop this aspect so much in the literature review.”
Authors’ answer:
Reviewer’s 1 consideration was duly noted and we decided no not develop this aspect so much in the literature review. Therefore, now it is possible to read: “It is believed that the voluntariness of temporary workers is directly linked with their professional outcomes. In fact, individuals who voluntarily choose to become temporary workers reveal higher levels of work satisfaction when compared with those whose choices were involuntary (Feldman, et al., 1995; Krausz, et al., 1995). As pointed out by Moorman and Harkand (2002), the underlying reason for taking a temporary work as-signment may indeed influence workers’ outcomes. According to this research, temporary workers who desire to gain experience or learn useful skills showed more positive organizational citizenship behavior than those whose only goal was to earn money (Moorman & Harkand, 2002).”
- 6. “The proportion of participants in the invited sample is surprisingly high (314-315). Perhaps it would be useful to include some reflection on the motivation of the subjects to participate in the study, especially considering the possible effect of the social desirability of the responses, bearing in mind that the invitation came from the company's management. It can be argued that the responses have a significant variance in the different dimensions.”
Authors’ answer:
We totally agree with Reviewer’s 1 consideration regarding this possible limitation. Therefore, in the current version of the manuscript, we explained in more detail how the data collection was carried out, giving particular emphasis to the information that was shared with the respondents. Thus, and to respond to this suggestion we introduce the following information in the "Sample” sub-section: “Respondents were assured that their answers were confidential and anonymous. Participants were also informed that they would have the opportunity to receive the overall results. These instructions were written on the questionnaire’s cover letter. The instructions explained that the questions were directly related to several parts of their daily work, specifically their perceptions of employment relationships. Participants were informed that the questionnaire was not a test and that there were no right or wrong answers. Workers were also assured that the companies would only have access to a final report and not to the raw data, as the data were used exclusively for academic research. The lead researcher’s email address was included in the cover letter in addition to a website address where respondents could find more information about the research project, including the in-volved academic organizations, its goals, outcomes, partners and other researchers included in the process. There was no incentive (cash or otherwise) for participating in this project. Because participation was voluntary and anonymous, participants did not sign an informed consent form.” Moreover, we acknowledge this possible limitation in the “Study Limitations” section by adding the following information: “Lastly, and because the invitation to be part of the study came from the temporary agencies’ management, the responses could be influenced by the social desirability effect. To prevent this possible limitation, we guaranteed that the respondents were fully aware of the project’s goals and that it was totally conducted by an independent institution. As a consequence, the results show a significant variance of the responses regarding the different variables and dimensions that were measured.”.
- 7. “It would be good if they could provide information on the fulfillment of the normality assumptions. If there were difficulties in this regard they could use boostrapping techniques. All this is not essential, only recommended.”
Authors’ answer:
Reviewer 1 correctly notes that information regarding the fulfillment of the normality assumptions was missing. In reply to this remark, we introduced the following: “The distribution of the motivation means across profiles fulfilled the normality assumption, with all the skewness values ranging between -3 and 3, and all the kurtosis values ranging between -10 and 10 (Kline, 2016).” and “(…) the distribution of the motivation means across profiles fulfilled the normality assumption, with all the skewness values ranging between -3 and 3, and all the kurtosis values ranging between -10 and 10 (Kline, 2016).”.
- “In line 357-358 two symbols are missing, I suppose chi2 and p-value, idem in 376 and 414”
Authors’ answer:
This is already corrected.
- “It is convenient to use 3 decimal places in places such as lines 358-359.”
Authors’ answer:
Although we understand the Reviewer 1 request. It is important to clarify that the manuscript formatting follows the APA 7º Edition rules. In accordance with such rules: numbers should be rounded consistently to two decimal places, unless the term is very small such as .001. APA 6.36. Therefore, we maintain the 2 decimal places unless the term is very small such as .001.
- “The CFAs are reported under two different sections, it is a bit confusing for the reader.”
Authors’ answer:
We appreciate this remark and agree that the presentation of the CFAs on two different sections can be confusing. Thus, we now present all the CFAs at the subsection “Confirmatory factor analysis”: “Our data analysis involved five phases. First, we performed two confirmatory factor analyses (CFA) to examine the factor structure of the items composing the affective commitment and the perception of HRP. These analyses where conducted with AMOS version 21.0 (Amos, Chicago, IL, USA). Regarding the affective commitment items, the overall goodness of the fit was based on combinations of several fit indices. The model had adequate fit to the data when there was a significant chi-square, .90 or higher for the Tucker Lewis (TLI) and fit indices (CFI), .06 or less for the root mean square error of approximation (RMSEA), and .08 or less for the standardized root mean square (SRMR) (Arbuckle, 2003). The model with one latent factor had good fit for the affective commitment with the agency (χ2 (7) = 182.00, r < .001; SRMR = .07; CFI = .99; TLI = .97; RMSEA = .08), and with the client company (χ2 (6) = 185.48, r < .001; SRMR = .06; CFI = .99; TLI = .97; RMSEA = .09). As to the HRP items, the model with one latent factor (i.e., HRP system) had good fit (χ2 (140) = 2134.57, r < .001; SRMR = .06; CFI = .96; TLI = .95; RMSEA = .06). The resulting 20-item scale had a reliability of .87, which is comparable to the one that Takeuchi et al. (2007) obtained for their HR system scale (.90).
The CFA conducted to evaluate discriminant validity and test for common method variance among the self-report measures revealed an acceptable fit for our four-factor theoretical model (χ2 (1008) = 9538,73, r >.001; SRMR = 0.06; CFI = .93; TLI = .92; RMSEA = .05). We compared this model with the one-factor model in which all items loaded on a single latent variable [χ2 (1019) = 26173,04, r < .001, SRMR = .11; CFI = .79, TLI = .77; RMSEA = .08]. Our theoretical model provided a better fit to the data [∆χ2 (11) = 16634,31, r < .001], indicating that the majority of variance in data cannot be explained by a single factor. We further tested an additional model (methods model) in which an unmeasured latent methods factor was added to the four-factor theoretical model. In this model, all items load on their theoretical constructs, as well as on the latent methods factor. The methods model obtained a good fit (χ2 (970) = 7109.73, r < .001; SRMR=.07; CFI = 0.95; TLI = 0.94; RMSEA = .04), and the method factor accounts for 13.6% of the variance, which falls below the threshold of 50% (Podsakoff et al., 2003). Although both models include the same observed variables, the methods model cannot be nested within the one-factor model, and for that reason we calculated the CFI difference to compare the goodness-of-fit of these models. The change of CFI between both models was .02, which is below the suggested rule of thumb of .05 (Bagozzi & Yi, 1990). Therefore, we conclude that common method bias is not a major concern in this study. Means, standard deviations, and correlations among the study variables are shown in Table 2, and the presence of distinct constructs can be observed.”.
- “In line with our predictions" would bring this type of commentary into the discussion, I would avoid it in the results section.”
Authors’ answer:
We totally agree with Reviewer’s 1 consideration and eliminated all this this type of commentary from the results presentation.
- 12. “Line 491: would use the diagonal of the table to incorporate these values.”
Authors’ answer:
After considering this commentary, we concluded that the alpha values were reported twice: they are presented when each of the scales is described, and also in the legend of the correlation table. Therefore, we eliminated this information from the legend of the correlation table.
- 13. “In the literature review and/or discussion I would perhaps add some reference to the different economic and labor regimes and work culture in which the reported studies were carried out, especially about the specificity of the Portuguese labor market of this particular study. Just if possible.”
Authors’ answer:
We appreciate this remark and agree that some reference to the specificity of the Portuguese labor market should be added. So, in “Study Limitations” section of the manuscript current version is it possible to read: “Second, the study focus is on Portuguese TAWs, and as result, extending and generalizing our findings to countries outside Europe should be done with considerable caution. But even if looking to these findings as a trend within the European Union, it is necessary to pay attention to the specificities of the Portuguese labor market. Chambel and Sobral (2019) pointed out three aspects that should be considered when analyzing the results: (1) the last economic crisis (before COVID 19), caused high levels of unemployment and re-stricted recruitment activities that limited the chances of obtaining a permanent contract; (2) Portuguese workers value job security and prefer contexts where they can find job sta-bility; and (3) since temporary agency work does not fulfill these values, the motives to choose this employment situation will be predominantly extrinsic. Even so, we believe that our study illustrates the importance of TAWs’ motivations in their employment rela-tionship. Future studies should approach this issue using representative samples and should consider a cross-cultural study.”
- 14. “It is very important to report the effect size values, eta squared for analysis of variance.”
Authors’ answer:
We totally agree with Reviewer’s 1 consideration and added that information in all analysis of variance: “The results of the ANOVAs conducted to compare motivation levels across profile groups revealed significant differences between intrinsic motivation (F = 3275.69, p < .001; η2 = .78), integrated motivation (F = 1479.12, p < .001; η2 = .78), identified motivation (F = 496.95, p < .001; η2 = .78), and extrinsic motivation (F = 416.44, p < .001; η2 = .78).” and “ANCOVA revealed significant differences between the profiles with regard to affective commitment toward the agency (F = 102.81, MSE = 1.91, p < .001; η2 = .11), affective commitment toward the client company (F = 54.69, MSE = 2.02, p < .001; η2 = .07), and perception of HRP (F = 98.26.62, MSE = 1.17, p < .001; η2 = .11).”
- 15. “It is very important to report Cohen's d for t-test (…) The use of t-test is announced (395) but I cannot find it in the results.”
Regarding this remark, we present our apologies to the Reviewers and to the Editors, but in fact, the reference to a “paired sample t-tests” is a mistake on our part. In fact, no t-test was performed, only Anovas and Ancovas. That is way no t-test results are presented. We are very sorry for this.
We would like to thank, for the Reviewer’s 1 comments, which helped to improve the paper considerably. We have been extremely careful to address each of the concerns and hope the Reviewer finds the revision fully responsive to his/her recommendations.
Reviewer 2 Report
Reviewed article has been prepared in accordance with applicable standards. The topic is current and important. The content of the article is consistent with the research problem defined in the title. The adopted formula of the title is not very scientific and does not sound good. I recommend to consider as follows: "The temporary agency worker's motivation profile analysis". In my opinion the quantity of literature positions is satisfying, but only a few have been published since 2015. It seems necessary to add some of the most recent entries.
Author Response
Reviewer’s 2 comments and questions:
Comments to the Author
- 1. “Reviewed article has been prepared in accordance with applicable standards. The topic is current and important. The content of the article is consistent with the research problem defined in the title.”
Authors’ answer:
We are pleased with the Reviewer’s overall positive evaluation of the paper. Thank you very much!
Specific items to consider:
2. “The adopted formula of the title is not very scientific and does not sound good. I recommend to consider as follows: "The temporary agency worker's motivation profile analysis".”
Authors’ answer:
We appreciate this remark and agree that title suggested by Reviewer 2 sounds “more scientific”. So, in the current version of the manuscript, we adopted the following title: "The temporary agency worker's motivation profile analysis"
- " In my opinion the quantity of literature positions is satisfying, but only a few have been published since 2015. It seems necessary to add some of the most recent entries.".”
Authors’ answer:
After considering this commentary, we added the following entries:
Barley, S.R., B.A. Bechky & F.J. Milliken (2017). The Changing Nature of Work: Careers, Identities, and Work Lives in the 21st Century. Academy of Management Discoveries, 3(2) 111–115. https://doi.org/10.5465/amd.2017.0034
Chambel, M. J., & Sobral, F. (2019). When temporary agency work is not so temporary. Economic and Industrial Democracy, 40(2), 238-256. https://doi.org/10.1177/0143831X18805931
Mas, A., & Pallais, A. (2020). Alternative work arrangements. Annual Review of Economics, 12, 631-658. https://doi.org/10.1146/annurev-economics-022020-032512
Organization for Economic Co-operation and Development (2019). OECD Employment Outlook 2019 - The Future of Work. OECD Publishing: Paris. https://doi.org/10.1787/9ee00155-en
Sobral, F., Chambel, M. J., & Castanheira, F. (2019a). Managing motivation in the contact center: The employment relationship of outsourcing and temporary agency workers. Economic and Industrial Democracy, 40(2), 357-381. https://doi.org/10.1177/0143831X16648386
Spreitzer, G. M., Cameron, L., & Garrett, L. (2017). Alternative work arrangements: Two images of the new world of work. Annual Review of Organizational Psychology and Organizational Behavior, 4, 473-499. https://doi.org/10.1146/annurev-orgpsych-032516-113332
These new entries can be found in the current version of the manuscript:
Line 29 and following:
“Today’s organization competitiveness depends on effectiveness, versatility, and ability to respond to customers (Flecker, al., 2009). As a consequence, flexibility (Barley, et al, 217; Spreitzer et al., 2017) along with the growth of contingent work (Organization for Economic Cooperation and Development, 2018, 2019) have become key concepts in work relations. It is increasingly common to find temporary workers side-by-side with those who are employed directly (Burgess & Connell, 2006; Spreitzer et al., 2017). It is therefore crucial for companies to understand how to engage these workers jointly (Zimmerman, et al., 2013). As mentioned by Mas and Pallais (2020), the equilibrium between the firm demands, the worker preferences and the different possibilities of work arrangements is still an open question.”
Line 225 and following:
“More recently, Sobral et al. (2019a) built a motivation profile typology of outsourcer and TAW in the contact center. Six profiles were identified: (a) involuntary motivation profile (i.e., high extrinsic and low intrinsic motivation; (b) moderate involuntary motivation (i.e., high extrinsic and identified motivation and medium integrated motivation); (c) low in-voluntary motivation (i.e., the extrinsic motivation was lower and closer to the integrated and identified motivation); (d) moderate voluntary motivation (i.e., high integrated and extrinsic motivation); (e) Voluntary Motivation (i.e., whit medium intrinsic and high extrinsic motivation); and (f) high motivation (i.e., a profile with high extrinsic and high autonomous motivations).”
Line 306 and following:
“Finally, Sobral et al. (2009a) results show that the motivation profile to which TAW and outsourcing workers belonged to, were able to differentiate their perceptions over the HRP and their affective commitment. In general, both outsourcing workers and TAW pro-files with a higher presence of intrinsic motivations presented better outcomes.”
Line 602 and following:
“We did not find a profile with low extrinsic and high intrinsic motivations, nor a profile with moderate scores on all types of motivation. In line with De Jong and Shalk (2010), and Sobral et al. (2019a) who reported the prevalence of external motivation in TAW, our results show a high presence of extrinsic motivation in the majority of profiles (with exception of the profile scoring low in all the motivations, i.e., the Low Motivation Profile).
More important, though, was the existence of intrinsic, integrated, identified, and extrinsic motivations inside the same profile, which reinforced the belief that different types of motivation are not necessarily exclusive and can co-exist with each other (e.g., González, et al., 2012; Sobral, et al. 2019a).”
Line 645 and following:
“As in the outcomes and construction of work motivation profiles of Moran et al. (2012) Sobral et al. (2019) and Van den Broeck et al. (2013), we also found a buffering effect, meaning that the presence of intrinsic motivations can attenuate the negative effect of the extrinsic motivations on the individual’s self-determination.”
Line 715 and following:
“But even if looking to these findings as a trend within the European Union, it is necessary to pay attention to the specificities of the Portuguese labor market. Chambel and Sobral (2019) pointed out three aspects that should be considered when analyzing the results: (1) the last economic crisis (before COVID 19), caused high levels of unemployment and restricted recruitment activities that limited the chances of obtaining a permanent contract; (2) Portuguese workers value job security and prefer contexts where they can find job stability; and (3) since temporary agency work does not fulfill these values, the motives to choose this employment situation will be predominantly extrinsic.”
Line 745 and following:
“In today’s globalized economy companies look for human flexibilization as a way of obtaining a competitive advantage (Spreitzer et al., 2017). In fact, the increasing use of contingent work in the last few decades can be considered as more than an economic matter; it can be considered as a reflex of on-going social change regarding temporary contingent contracts (Mas & Pallais, 2020). The introduction and growth of labor market flexibility has brought new challenges to HR management.”
We would like to thank, for the Reviewer’s 1 comments, which helped to improve the paper considerably. We have been extremely careful to address each of the concerns and hope the Reviewer finds the revision fully responsive to his/her recommendations.